

# An experimental study on the performance of collaborative filtering based on user reviews for large-scale datasets

Sumaia AL-Ghuribi[1,2], Shahrul Azman Mohd Noah[1] and Mawal Mohammed[3]

[1] Center for Artificial Intelligence Technology, Universiti Kebangsaan Malaysia, Bangi, Selangor, Malaysia
[2] Department of Computer Science, Faculty of Applied Sciences, Taiz University, Taiz, Yemen
[3] Department of Software Engineering, Prince Sattam Bin Abdulaziz University, Alkharj, Saudi Arabia

## ABSTRACT

Collaborative filtering (CF) approaches generate user recommendations based on user similarities. These similarities are calculated based on the overall (explicit) user ratings. However, in some domains, such ratings may be sparse or unavailable. User reviews can play a significant role in such cases, as implicit ratings can be derived from the reviews using sentiment analysis, a natural language processing technique. However, most current studies calculate the implicit ratings by simply aggregating the scores of all sentiment words appearing in reviews and, thus, ignoring the elements of sentiment degrees and aspects of user reviews. This study addresses this issue by calculating the implicit rating differently, leveraging the rich information in user reviews by using both sentiment words and aspect–sentiment word pairs to enhance the CF performance. It proposes four methods to calculate the implicit ratings on large-scale datasets: the first considers the degree of sentiment words, while the second exploits the aspects by extracting aspect-sentiment word pairs to calculate the implicit ratings. The remaining two methods combine explicit ratings with the implicit ratings generated by the first two methods. The generated ratings are then incorporated into different CF rating prediction algorithms to evaluate their effectiveness in enhancing the CF performance. Evaluative experiments of the proposed methods are conducted on two large-scale datasets: Amazon and Yelp. Results of the experiments show that the proposed ratings improved the accuracy of CF rating prediction algorithms and outperformed the explicit ratings in terms of three predictive accuracy metrics.

Corresponding author
Shahrul Azman Mohd Noah,
shahrul@ukm.edu.my

# INTRODUCTION

Recommender systems (RSs) aim to help users discover relevant items based on their preferences. Collaborative filtering (CF) is among the most widely applied approaches to recommendations (*Jamil, Noah & Mohd, 2020*; *Alhijawi et al., 2021*). CF generates recommendations for a target user based on the similarities with other users who have previously shown similar preferences or interests (*Aciar et al., 2007*). Most CF techniques only include one factor when measuring similarities–the overall ratings (also known as explicit ratings), representing the users' overall opinion about the items (*Zhang et al., 2013*).

However, the inadequacy of such ratings has exposed CF-based approaches to problems associated with data sparsity which may harm the recommendation accuracy (*AL-Ghuribi & Noah, 2019*; *Osman et al., 2021*). A few studies used different types of user-generated information, such as tags (*Ghabayen & Noah, 2014*; *Pan et al., 2021*; *Ge et al., 2015*) and social relationships (*Beilin & Yi, 2013*; *Chen et al., 2013*; *Shokeen, Rana & Rani, 2021*), to solve problems associated with data sparsity. However, these methods are still inadequate, especially when historical data about the target user is insufficient.

Social media and e-commerce sites encourage users to provide reviews describing their assessment of items (*Murshed et al., 2022*). These reviews are unquestionably valuable sources to help identify user preferences for specific items. Various elements of reviews, such as the topics discussed, the multi-faceted nature of opinions, contextual information, comparative opinions, and reviewers' emotions, can be unravelled and analyzed using information extraction and text analytic techniques (*AL-Ghuribi & Noah, 2019*; *Chen, Chen & Wang, 2015*). Despite the rich information in user reviews, few studies have explored their utility as a tool for improving recommendation accuracy and addressing CF issues (*Osman et al., 2021*; *Pappas & Popescu-Belis, 2016*; *Hasanzadeh, Fakhrahmad & Taheri, 2022*).

Current studies focusing on inferring ratings from reviews use only a single method that relies solely on the sentiment words included in the review, such as the works of *Zhang et al. (2013)*, *Osman et al. (2021)*, *Pappas & Popescu-Belis (2016)*, *García-Cumbreras, Montejo-Ráez & Díaz-Galiano (2013)* and *Rafailidis & Crestani (2019)*. They relied on the sentiment classification method to classify the review into three categories: negative, neutral, and positive (*Zhang et al., 2013*; *Osman et al., 2021*; *García-Cumbreras, Montejo-Ráez & Díaz-Galiano, 2013*). In this case, a review is classified as positive if positive sentiment terms are more dominant than negatives. The degree of the sentiment orientation of the terms is not emphasized, even though words differ in their strength in expressing opinions. For example, "great" has a stronger sentiment orientation than "good". As CF relies on the quality and accuracy of ratings, ignoring the orientation of the sentiment terms will affect CF performance in terms of rating prediction and recommendation.

Few researchers (*Da'u et al., 2020*; *Ray, Garain & Sarkar, 2021*; *AL-Ghuribi et al., 2023*) have also considered using aspects to improve CF recommendations, as the aspects can provide better information about user preferences. However, the current studies that utilize aspects mainly concentrate on extracting aspects to represent user preferences instead of figuring out their implicit ratings (*Ray, Garain & Sarkar, 2021*; *Akhtar et al., 2017*; *Liu, Zhang & Gulla, 2021*). For example, the work of *Akhtar et al. (2017)* classified aspects in the TripAdvisor hotel dataset into predefined classes such as 'value', 'location', and 'service'. Based on reviews belonging to each class, a sentiment polarity of positive, negative, and neutral will be assigned to the category. The work of *Ray, Garain & Sarkar (2021)* followed a similar concept of using aspects but used the Bidirectional Encoder Representations from Transformers (BERT) model to classify the aspects based on the queries provided by users. *Liu, Zhang & Gulla (2021)* proposed a multilingual review-based recommender system based on aspect-based sentiment analysis. Using the word-embedding technique

for building the aspect-embedding model, a proposed aspect-based sentiment prediction module will generate a kind of document-level sentiment distribution of the reviews.

This study addressed the abovementioned issues by proposing two types of implicit ratings. The first is by considering the degree of sentiment words to calculate the implicit ratings. The second exploits the aspects to calculate the implicit ratings by extracting the aspect-sentiment word pairs. We also assess the impact of the generated implicit ratings when combined with the explicit ones. The combination of implicit and explicit ratings was based on our assumption that explicit ratings that express users' overall opinions of the items are seen as having the potential to enhance the rating prediction performance if combined with the generated implicit ratings. As a result, during the experiments, four variants of ratings were proposed and subsequently experimented, as follows.

- Implicit ratings using the degree of sentiment words (*ImplicitRating_SW*)
- Implicit ratings using aspects (*ImplicitRating_Aspect*)
- Combination of explicit rating with *ImplicitRating_SW (AVG_ImpSW)*
- Combination of explicit rating with *ImplicitRating_Aspect (AVG_ImpAspect)*

In summary, this study presents review-based CF approaches that employ four alternative implicit rating variants extracted from user reviews. To our knowledge, it is the first study to demonstrate various methods for producing implicit ratings on large-scale datasets. All the experiments applied to develop the proposed approaches are conducted on two real large-scale datasets: Amazon and Yelp. The novelty of this study is that it focuses on calculating the implicit rating in a different manner that leverages the rich information in user reviews by using both sentiment words and aspect–sentiment word pairs to enhance the CF performance and mitigate its problems. The impact of the proposed ratings is evaluated against the baseline rating (*i.e.*, explicit rating) by comparing the performance of the CF rating prediction algorithms using the explicit and proposed ratings.

The rest of the article is organized as follows. Section 2 briefly reviews the state of the art in CF and sentiment analysis. Next, Section 3 describes the methodology we adopted to develop our proposed approach. Section 4 presents the experiments we conducted using the review-based CF approaches, followed by an evaluation in Section 5. Finally, Section 6 concludes the study.

## State of the art

RSs have been used on several platforms, such as e-commerce and social networks. RSs use three main approaches to create recommendations: content-based, collaboration-based, and hybrid. RSs typically focus on two issues: predicting ratings and recommending items (*Ricci, Rokach & Shapira, 2015*; *Fayyaz et al., 2020*). This study focuses on rating predictions for the CF approach, the most popular approach used in RSs (*Yang et al., 2016*; *Nguyen & Amer, 2019*).

### Collaborative filtering

CF generates a recommendation for a user based on the similarities among users who had similar preferences/interests in the past. This approach assumes that individuals with similar preferences in the past are likely to have similar preferences in the future (*Aciar et*

*al., 2007*). CF identifies new user-item associations by determining user relationships and the interdependencies between items (*Yang et al., 2016*). Generally, a CF model consists of a set **U** of users and their preferences over a set of item **I**. Then, a utility function **R** measures the suitability of recommending item $i \in I$ to user $u \in U$. It is defined as **R**: $U \times I \rightarrow R_0$, where $r \in R_0$ is either a real number or a positive integer within a specific range (*Adomavicius, Manouselis & Kwon, 2011*).

CF can be grouped into two categories: memory-based and model-based (*Chen, Chen & Wang, 2015*). Memory-based CF can be motivated by the observation that users usually prefer recommendations from like-minded consumers. These methods apply nearest-neighbour-like algorithms to predict a user's ratings based on the ratings given by like-minded users. These algorithms can be classified into user- and item-based methods (*Adomavicius & Tuzhilin, 2005*). The former identifies a set of neighbours for a target user and then recommends a set of items based on the neighbours' interests. In contrast, the latter recommends items similar to (share features with) items that the user has previously purchased, viewed, or liked. The three most frequently used metrics/measures to identify user/item similarities are the Pearson correlation, Euclidean distance and Cosine-based similarity (*Liu, Mehandjiev & Xu, 2011*; *Amer, Abdalla & Nguyen, 2021*). On the other hand, the model-based CF predicts users' ratings of unseen items by developing a descriptive model of user preferences and then using it for predicting ratings. Many of these methods are inspired by machine learning algorithms.

Generally, the performance of the CF approach depends on the availability of sufficient user ratings (*Aciar et al., 2007*). Such a dependency, however, may cause CF to suffer from sparsity, cold start, scalability, and rating bias problems (*AL-Ghuribi & Noah, 2019*; *García-Cumbreras, Montejo-Ráez & Díaz-Galiano, 2013*). The sparsity issue has attracted much attention, and various approaches have been proposed to solve it. One of the approaches is to extract valuable information from user reviews using sentiment analysis techniques and integrate them into CF (*Chen, Chen & Wang, 2015*). The following section provides a brief overview of sentiment analysis.

## Sentiment analysis

Sentiment analysis or opinion mining is the computational study of people's opinions, sentiments, emotions, appraisals, and attitudes towards entities such as products, services, organizations, individuals, issues, events, topics, and attributes (*Liu, 2012*). Approaches to sentiment analysis may involve various fields, such as natural language processing (NLP), information retrieval and machine learning. Sentiment analysis can be applied at three primary levels: the document level, the sentence level, and the aspect level, and the approaches can be roughly divided into two categories: machine learning and lexicon-based approaches. Both aim to determine or classify the user sentiments associated with each entity in the given text. Generally, most document-level and sentence-level sentiment classification methods rely on machine-learning approaches. In contrast, aspect-based sentiment classification methods are mostly lexicon-based approaches. The machine learning approach mainly involves extracting and selecting a proper set of features to detect opinions (*Liu, 2012*). Lexicon-based approaches primarily depend on a sentiment lexicon,

which is a collection of precompiled known sentiment words (*Serrano-Guerrero et al., 2015*). Lexicon-based approaches can be either dictionary-based or corpus-based (*Darwich, Noah & Omar, 2020*).

## Incorporating sentiment analysis into collaborative filtering algorithms

CF approaches rely on ratings and cannot function effectively without enough ratings. However, user reviews can serve as an alternative source in such cases by allowing sentiment analysis techniques to generate the overall rating. *Leung, Chan & Chung (2006)* were among the first to point out the possible benefits of combining sentiment analysis with CF approaches to increase accuracy by inferring ratings from user reviews when explicit ratings are unavailable. Following *Leung, Chan & Chung (2006)*, *Aciar et al. (2007)* were the first study that attempted to use user reviews to build RSs by developing an ontology to transform review content into a standardized form that could be used to provide recommendations. More studies have proposed ways to integrate user reviews into CF to provide better recommendations with different combinations of other elements or enhanced with other models.

*García-Cumbreras, Montejo-Ráez & Díaz-Galiano (2013)* proposed an approach that classifies users into two classes (Pessimist and Optimist) according to the average polarity of the user reviews. The classes are included as a new attribute to the CF algorithm. Evaluation using a newly created corpus from the Internet Movie Database (IMDb) shows the enhanced the CF performance.

*Zhang et al. (2013)* proposed an algorithm to calculate an overall rating (called the "virtual rating") from user reviews by aggregating the sentiment words with the reviews' emoticons. They proposed a self-supervised sentiment classification approach involving both lexicon-based and corpus-based models that could determine the overall sentiment polarity of reviews containing textual words and emoticons.

*Wang & Wang (2015)* proposed an approach that mapped extracted opinions from online reviews into preferences in a way that CF-based RSs could understand. The experimental results for their proposed CF approach indicated that incorporating sentiment analysis into CF improves the accuracy of product recommendations.

*Pappas & Popescu-Belis (2016)* consider the user reviews to address the problem of one-class CF problem, which only handles explicit positive feedback. Extracted sentiment information from reviews is mapped as sentiment scores to user ratings using an integrated nearest neighbour model. Their proposed approach demonstrated that users' unary feedback (likes or favourites) and the sentiments expressed in user reviews are inextricably linked.

*Dubey et al. (2018)* proposed an enhanced item-based CF approach using sentiment analysis. They built a dictionary, calculated the sentiment scores by the positive reviews' probability, and filtered out items with negative reviews. The proposed approach improved the quality of predictions compared to conventional RSs. The limitation of this work is that the authors claim that their method of calculating the sentiment score based on probabilities is inaccurate because many reviews are classified incorrectly.

*Ghasemi & Momtazi (2021)* proposed a review-based model to boost CF by measuring user similarities based on seven different methods, including two that are lexicon-based, two that use neural representations of words, and three that are based on neural representations of text. Among these, the model based on a long short-term memory (LSTM) network achieved the best results.

Even though previous studies have found that sentiment analysis benefits the performance of the CF approach, the existing studies are still inadequate. Most of these studies calculate the implicit ratings using the conventional method that aggregates sentiment scores in a review or employing a classification method based on most positive or negative words. Both methods ignore the degree of sentiment words during calculating the implicit rating, which is not the case in this study. Furthermore, most recent studies that attempt to determine implicit ratings rely only on the sentiment words appearing in user reviews. In contrast, our study calculates the implicit rating using aspect–sentiment word pairs in addition to the method of aggregating sentiment words considering their degrees.

Other approaches that use user reviews go even further by proposing sentiment-based models that integrate contextual information to enhance the CF (*Osman et al., 2021*) or generate user profiles to identify user preferences (*Cheng et al., 2019*; *Bauman, Liu & Tuzhilin, 2017*; *Wang, Wang & Xu, 2018*). Moreover, others like (*Da'u et al., 2020*; *Ray, Garain & Sarkar, 2021*; *AL-Ghuribi et al., 2023*; *Cheng et al., 2018*) focused on integrating aspect/feature elements of reviews into CF as multiple criteria to improve the CF performance. As can be seen from the current studies, little emphasis has been placed on using user reviews to determine the implicit rating and integrate it into the CF approach. This motivated us to use user reviews to calculate implicit ratings using different methods that are not previously indicated in the available studies.

Some researchers also claim that reviews are insufficient for acquiring user-personalized information and that information stored in the user-item rating matrix is needed because different users may write identical reviews but assign different ratings to the same item (*Wang et al., 2019*). Therefore, the present study proposes two rating variants that combine the explicit ratings from the rating matrix with different implicit ratings derived from user reviews to determine whether user reviews are sufficient for producing personalized recommendations or require explicit ratings. Finally, the present study focuses on a large-scale dataset, which is not the case in many previous studies, and applying those methodologies from past studies to a large-scale dataset could result in imprecise ratings and negatively influence rating predictions (*Hasanzadeh, Fakhrahmad & Taheri, 2022*).

To summarise, we proposed four variants of implicit rating in this study. The four variants of implicit ratings were integrated into various CF rating prediction algorithms and compared against the overall (explicit) ratings. The aims are twofold. The first is to assess the implicit ratings' effectiveness in enhancing recommendation algorithms' prediction accuracy. The second is to determine the influence of implicit ratings on the performance of prediction accuracy when combined with explicit ratings. The ideas behind these aims are due to the distinctions between the overall and user reviews-derived ratings. Overall ratings are relatively objective, focusing on general preference or satisfaction.

They do not capture the reasons behind the rating, the specific aspects that influenced the rating, or the contextual factors that shaped the user's evaluation. User reviews, on the hand, capture the subjective nature of opinions. Users can express their preferences, perspectives, and subjective experiences in the review text. They provide context-specific insights and can highlight the factors that influence the user's evaluation, such as specific features, quality, customer service, or price. Thus, the question of whether combining general preferences with the subjective nature of opinions can produce better prediction accuracy will be addressed by the experiment conducted in this study.

## Method

This study aims to examine the effects of implicit ratings generated from user reviews compared to explicit ratings regarding the accuracy of CF rating prediction algorithms. We consider two variants of implicit ratings: ratings based on extracted sentiment word scores and ratings based on extracted aspect–sentiment word pairs. Apart from that, we also investigate the possibility of proposing new ratings by combining both implicit and explicit ratings. As such, the general methodology of the experimental study involved three tasks, as shown in Fig. 1.

- Task 1: Calculate two variants of the implicit rating extracted from user reviews.
- Task 2: Produce two additional variants for the overall rating:

  – The first variant combines the explicit and implicit ratings based on extracted sentiment word scores.
  – The second variant combines the explicit and implicit ratings based on aspect–sentiment word pair scores.

- Task 3: Integrate the generated ratings into various CF rating prediction algorithms and evaluate their accuracy compared to the explicit ratings.

  The following subsections describe these tasks in detail.

## Task 1: Generating implicit rating

This task aims to compute an implicit rating, which is the total review sentiment score, determined by analyzing users' reviews that reflect their opinions of the consumed item. There are two methods for calculating implicit rating, based on either sentiment words or aspect–sentiment word pairs. Descriptions of the two methods follow.

## Implicit rating using sentiment words

This method involves extracting all sentiment words mentioned in a review and calculating their scores based on a specific lexicon. The overall implicit rating is the sum of the scores of all extracted sentiment words ($SW$) mentioned in review $i$:

$$Implicit\ Rating\_SW_i = \sum_{j=1}^{N} Score\left(SW_j\right) \qquad (1)$$

where $N$ is the number of sentiment words in review $i$ and $Score(SW_j)$ is the sentiment score for the sentiment word $j$ mentioned in the review.

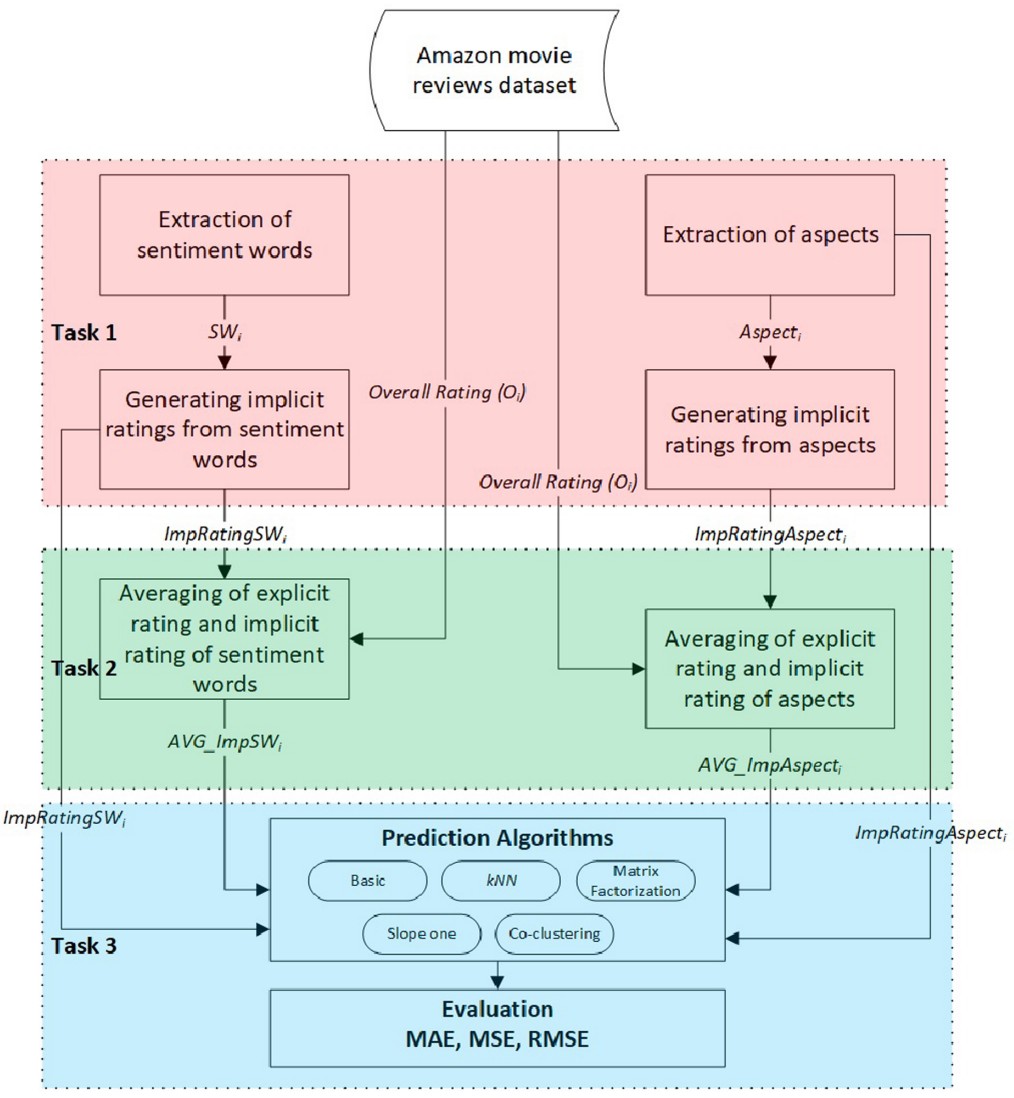

**Figure 1  The general methodology of the experimental study.**

The selection of sentiment words and the assignment of their scores are based on our earlier work presented in *AL-Ghuribi, Noah & Tiun (2020a)*. In this work, the sentiment words were selected from four parts of speech: noun, verb, adjective, and adverb because these parts of speech contain sentiment that can considerably influence the sentiment analysis process. The work (*AL-Ghuribi, Noah & Tiun, 2020a*) also provided a domain-based lexicon containing numerous sentiment words with their scores, which are used for assigning the score of *Score* ($SW_j$).

## Implicit rating using aspects

The second method for calculating implicit ratings relies on aspect–sentiment word pairs. It is based on the hypothesis that the implicit rating is the weighted sum of the user's opinions (*i.e.*, ratings) about multiple aspects. Many experiments have shown this hypothesis to be

reliable (*Ngoc, Thu & Nguyen, 2019*; *Yu et al., 2011*; *Wang, Lu & Zhai, 2010*; *Al-Ghuribi, Noah & Tiun, 2020b*).

To achieve this, the aspects are first extracted from reviews using the Semantically Enhanced Aspect Extraction (SEAE) method proposed in *Al-Ghuribi, Noah & Tiun (2020b)* and the blocking technique mentioned in *Thabit & AL-Ghuribi (2013)*. Then each aspect's weight is determined using the Modified Term Frequency–Inverse Document Frequency (TF-IDF) proposed by *Zhu, Wang & Zou (2016)*. Finally, the domain-specific lexicon proposed in *AL-Ghuribi, Noah & Tiun (2020a)* is used to determine the scores for the sentiment words corresponding to each aspect. The implicit rating produced by this method is the weighted sum of the ratings of sentiment words belonging to each aspect mentioned in a review, calculated as follows:

$$Implicit\ Rating\_Aspect_i = \sum_{j=1}^{k} R_j W_j \tag{2}$$

where $k$ is the number of aspects in review $i$, $R_j$ and $W_j$ refer to the sentiment word's rating and weight for aspect $j$ in review $i$.

We provide an example from the Amazon movie dataset to further explain the difference between the two generated implicit ratings. Assume that a user gives the following review:

"*This is a charming version of the classic Dickens tale. Henry Winkler makes a good showing as the "Scrooge" character. The casting is excellent and the music old but very relevant.*"

According to the first method based on extracted sentiment word scores, the implicit rating is 1.1075, as shown in Table 1.

The second method, which is based on aspect-sentiment word pairs, yields an implicit rating of 0.1202. This rating is obtained as shown in Table 2, where each aspect's weight is assigned based on the modified TF-IDF proposed in *Zhu, Wang & Zou (2016)*, and the sentiment scores are assigned based on the domain-specific lexicon proposed in *AL-Ghuribi, Noah & Tiun (2020a)*.

## Task 2: Produce two variants for the overall rating

This task generates two new variants for the overall rating by combining the implicit rating produced in Task 1 with the explicit rating, $O$. These ratings are calculated as the average of the explicit and the implicit ratings. Thus, one of the new variants is the average of the explicit and implicit ratings using sentiment words for the given review, as given in Eq. (3). The other is the average rating of the explicit and the implicit ratings using aspect–sentiment word pairs for the given review, as shown in Eq. (4).

$$AVG\_ImpSW_i = \frac{\sum_{j=1}^{N} Score\left(SW_j\right) + O_i}{2} \tag{3}$$

$$AVG\_ImpAspect_i = \frac{\sum_{j=1}^{k} R_j W_j + O_i}{2} \tag{4}$$

**Table 1 An example of generating an implicit rating using sentiment words for the given review (indicating a positive review).**

| Sentiment word | Score | Sentiment word | Score |
|---|---|---|---|
| charming | 0.0310 | showing | 0.0098 |
| version | −0.0097 | scrooge | 0.0110 |
| classic | 0.2444 | character | 0.0089 |
| dickens | 0.0094 | casting | 0.0111 |
| tale | 0.0385 | excellent | 0.2640 |
| henry | 0.0176 | music | 0.0890 |
| winkler | 0.0030 | old | 0.1105 |
| makes | 0.0705 | relevant | 0.0107 |
| good | 0.1877 | | |
| **Total review sentiment score** | | | **1.1075** |

**Table 2 An example of generating an implicit rating using aspect-sentiment word pairs for the given review (indicating a positive review).**

| Aspect-sentiment word pairs | Aspect weight | Sentiment scores | Aspect-sentiment pair's score |
|---|---|---|---|
| (version, charming) | 0.3801 | 0.0310 | 0.01178 |
| (tale, classic) | 0.1332 | 0.2444 | 0.03255 |
| (showing, makes_good) | 0.0736 | 0.2582 | 0.01900 |
| (casting, excellent) | 0.0703 | 0.2640 | 0.01856 |
| (music, old) | 0.3163 | 0.1105 | 0.03495 |
| (music, very_relevant) | 0.3163 | 0.0107 | 0.00338 |
| **Total review aspect–sentiment score** | | | **0.12022** |

## Task 3: Evaluation of the proposed ratings

This study aims to improve CF performance in terms of rating prediction when ratings are derived from user reviews. In the first two tasks, four different ratings are generated. This task evaluates the effectiveness of these ratings in improving the performance of CF rating prediction algorithms by integrating them independently into various CF rating prediction algorithms. This study uses the rating prediction algorithms available in the Surprise library (*Hug, 2020b*). This library offers a variety of benchmark rating prediction algorithms dedicated to RSs. These algorithms range in complexity from the most basic to the most advanced. Table 3 lists all the algorithms used in this study.

As can be seen, there are five categories for rating prediction algorithms, starting with the most fundamental, like BaselineOnly and NormalPredictor. Follows by the $k$ nearest neighbour ($k$NN) algorithms, which are mainly used for regression and classification (*Abdalla & Amer, 2021*; *Noaman & Al-Ghuribi, 2012*; *Al-Ghuribi & Alshomrani, 2013*), and have evolved as one of the most popular fundamental algorithms for CF recommendation systems. There are four different $k$NN algorithms: the basic algorithm ($k$NNBasic), the algorithm that considers the mean ratings of each user ($k$NNWithMean), the algorithm that considers the normalization of each user's z-score ($k$NNWithZScore), and the basic algorithm that considers a baseline rating ($k$NNBaseline).

**Table 3  The prediction algorithms used.**

| Category of algorithms | Algorithm |
|---|---|
| Basic algorithms | NormalPredictor |
| | BaselineOnly |
| *k* nearest neighbor algorithms | *k*NNBasic |
| | *k*NNWithMeans |
| | *k*NNWithZScore |
| | *k*NNBaseline |
| Matrix factorisation algorithms | SVD |
| | SVDpp |
| | NMF |
| Slope One | SlopeOne |
| Co-clustering | Co-clustering |

Three algorithms fall under the category of "matrix factorization": Singular Value Decomposition (SVD), another version of SVD that takes into account implicit ratings (SVDpp), and an algorithm based on Non-negative Matrix Factorization (NMF). The last two algorithms are SlopeOne, a simple implementation of the SlopeOne algorithm, and Co-clustering, based on the co-clustering technique. It is beyond the scope of this article to describe the details of all these algorithms; further descriptions are available in *Hug (2020a)*.

Simply put, in this task, the proposed ratings are independently integrated into the 11 rating prediction algorithms listed in Table 3 to evaluate their impacts on improving the CF performance.

Evaluation of the results is based on three predictive accuracy metrics: mean absolute error (MAE), mean square error (MSE), and root mean square error (RMSE) metrics. A lower value of these metrics indicates a higher CF performance because they calculate the difference between predicted and actual ratings (*Al-Ghuribi & Noah, 2021*). The equations of the three metrics are as follows.

$$\text{MAE} = \frac{\sum_{i=1}^{S}(p_i - r_i)}{S} \tag{5}$$

$$\text{MSE} = \frac{\sum_{i=1}^{S}(p_i - r_i)^2}{S} \tag{6}$$

$$\text{RMSE} = \sqrt{\frac{\sum_{i=1}^{S}(p_i - r_i)^2}{S}} \tag{7}$$

where $S$ is the size of the test set, $p_i$ is the predicted rating calculated by the CF approach, and $r_i$ is the actual rating given by the user.

This study is the first to demonstrate various methods for generating implicit ratings on large-scale datasets. As a result, the explicit rating provided in the dataset is used as

the baseline rating. Most of the available studies develop various CF models using explicit ratings. Specifically, the performance of each rating prediction algorithm mentioned in Table 3 using the proposed ratings is evaluated against the performance of each algorithm using the baseline rating (*i.e.*, the explicit rating). The purpose of using the explicit ratings is to see whether the proposed ratings will outperform the explicit rating in enhancing the CF performance.

# EXPERIMENTAL RESULTS AND ANALYSIS

## The Dataset

The proposed review-based CF approaches are evaluated using the movie domain of the Amazon dataset (http://jmcauley.ucsd.edu/data/amazon/links.html) (*McAuley & Leskovec, 2013*) and the restaurant domain of the Yelp dataset (https://www.yelp.com/dataset). Both datasets are considered as enormous datasets used to assess RSs (*Seo et al., 2017*). The movie and restaurant domains are the most widely used and popular in recent studies (*Da'u et al., 2020*; *Al-Ghuribi & Noah, 2021*; *Dridi, Tamine & Slimani, 2022*; *Nawi, Noah & Zakaria, 2021*). Figure 2 shows an example of the Amazon and Yelp datasets. Each review in both datasets is rated in the range of one and five, with one and two ratings denoting negative reviews, three denoting neutral reviews, and four and five ratings denoting positive reviews (*Labille, Alfarhood & Gauch, 2016*). Table 4 describes the datasets used in this study, and Fig. 3 shows the rating distribution for both datasets. As can be observed, over 800,000 movies in the Amazon dataset and over 400,000 restaurants in the Yelp dataset received five-star reviews.

## The experiment setting

Three parameters must be identified before the three tasks of this study can be carried out. The parameters are as follows:

- $\lambda$–The minimum number of rated items (movies/restaurants) per user and the minimum number of ratings for each item (movie/restaurant).
- $\xi$–The rating scale is the standard rating scale to which all generated ratings are rescaled from their original rating scales.
- $k$–The optimal number of neighbours that will be used in all $k$NN rating prediction algorithms.

Different experiments are conducted to determine the optimal value for each parameter, with each experiment utilizing five-fold cross-validation. In each fold, the dataset is divided into training and testing (80% and 20%, respectively). The following subsections describe the conducted experiments.

### The minimum number of rated items per user and ratings for each item ($\lambda$)

As indicated in Table 4, the Amazon dataset has 123,960 users, 1,697,471 reviews, and 50,052 movies, whereas the Yelp dataset has 401,867 users, 1,000,000 reviews, and 145,636 restaurants. The parameter $\lambda$ is meant to choose users who have written enough reviews to ensure that the proposed approaches work effectively. It is also used to select

```
{
  "reviewerID": "A2SUAM1J3GNN3B",
  "asin": "0000013714",
  "reviewerName": "J. McDonald",
  "helpful": [2, 3],
  "reviewText": "I bought this for my husband who
plays the piano.  He is having a wonderful time
playing these old hymns.  The music  is at times
hard to read because we think the book was
published for singing from more than playing
from.  Great purchase though!",
  "overall": 5.0,
  "summary": "Heavenly Highway Hymns",
  "unixReviewTime": 1252800000,
  "reviewTime": "09 13, 2009"
}
```
```
{
  " reviewerID" : "jsDu6QEJHbwP2Blom1PLCA",
  " businessID" : "Ums3gaP2qM3W1XcA5r6SsQ",
  " reviewerName":"msQe1u7Z_XuqjGoqhB0J5g ",
  " helpful": [0],
  "reviewText":"Delicious healthy food. The steak is
amazing. Fish and pork are awesome too. Service
is above and beyond. Not a bad thing to say about
this place. Worth every penny!',
  " stars": [5],
  " reviewTime":" 9/5/2014 ",
  " Funnny": [0],
  " cool": [0]
}
```

(a)   5-core Amazon dataset                (b)   Yelp dataset

**Figure 2**   **Sample of a record from the used datasets.**

**Table 4**   **Details of the used datasets.**

| Attribute | Amazon | Yelp |
| --- | --- | --- |
| Number of records | 1,697,471 | 1,000,000 |
| Number of unique product IDs | 50,052 | 145,636 |
| Number of unique users | 123,960 | 401,867 |
| Average number of ratings per user | 13.69 | 2.49 |
| Average number of ratings per product | 33.91 | 6.87 |

movies/restaurants with a minimum number of reviews. For example, $\lambda = 10$ selects users who have rated at least ten movies/restaurants and selects movies/restaurants with at least ten users who have rated. For testing purposes, we chose five different values for this parameter: 10, 15, 20, 30, and 50. Table 5 shows the number of reviews, users, and items for each $\lambda$ value for both datasets.

Different experiments for the CF rating prediction process using the Surprise library (*Hug, 2020b*) are conducted to select the optimal value for the $\lambda$ parameter. Each of the 11 rating prediction algorithms listed in Table 3 was applied to the five sub-datasets determined by the five possible values of $\lambda$, thus yielding 55 results for each dataset. All experiments used the overall rating (*i.e.*, the explicit or baseline rating) provided in the dataset.

The MAE, MSE, and RMSE are used to evaluate the rating prediction accuracy of each algorithm with different $\lambda$ values. Figure 4 illustrates the results of these metrics for the Amazon dataset.

The figure demonstrates that for all metrics, the dataset size has no impact on the order of prediction accuracies for the rating prediction algorithms (*i.e.*, highest accuracy to the lowest accuracy). Mainly, for the five values of $\lambda$, the SVDpp algorithm provides the best results for the three metrics, followed by the BaselineOnly algorithm. On the other hand, the NormalPredictor algorithm gives the lowest accuracy, followed by the *k*NNBasic algorithm. The Yelp dataset shows a similar trend. Therefore, we chose $\lambda = 20$ for the
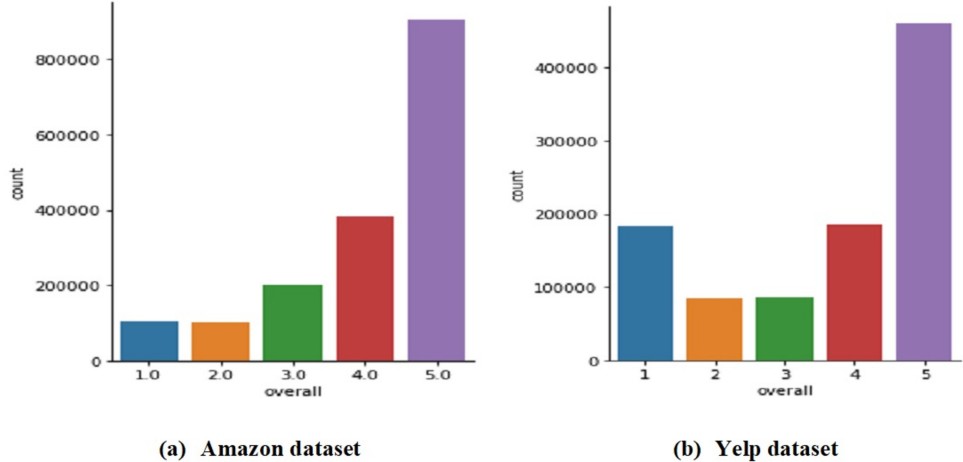

(a) **Amazon dataset**                    (b) **Yelp dataset**

**Figure 3   Distribution of the ratings in the used datasets.**

**Table 5   Dataset sizes based on different values of the parameter λ.**

| Value of λ | Number of records/reviews | Number of users | Number of movies |
| --- | --- | --- | --- |
| 10 | 1,016,863 | 35,106 | 28,620 |
| 15 | 787,512 | 19,708 | 21,435 |
| 20 | 650,145 | 13,214 | 17,022 |
| 30 | 482,615 | 7,426 | 11,908 |
| 50 | 319,524 | 3,684 | 7,251 |
| Amazon dataset | | | |

| Value of λ | Number of records/reviews | Number of users | Number of restaurants |
| --- | --- | --- | --- |
| 10 | 162,723 | 11,831 | 20,358 |
| 15 | 94,823 | 5,727 | 12,950 |
| 20 | 59,949 | 3,275 | 8,973 |
| 30 | 29,248 | 1,458 | 5,038 |
| 50 | 9,985 | 527 | 2,155 |
| Yelp dataset | | | |

Amazon dataset and $\lambda = 10$ for the Yelp dataset to obtain pertinent data (*i.e.*, a fair number of records/reviews, users, and movies/restaurants).

### The rating scale (ξ)

In this study, we proposed four different ratings, each of which includes a rating scale. The rating scales for these ratings as well as the explicit rating scale, are listed in Table 6. It is clear that each rating has a different scale from the explicit rating scale. Hence, standardizing the scales to a single, uniform scale is required before their usage in CF systems. The standardization can be achieved through a normalization process that rescales the data from an original range to a new range. In this case, transforming the generating ratings by scaling them to the same scale as the explicit rating (*i.e.*, [1, 5]). There are several
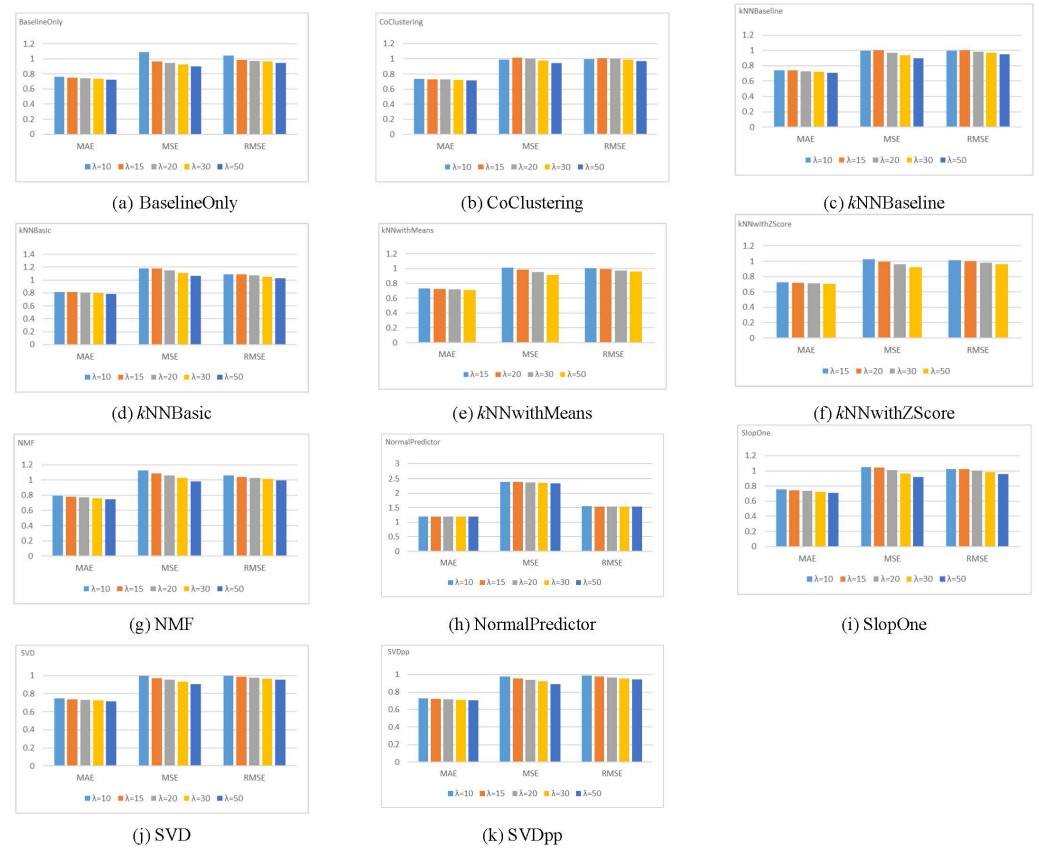

**Figure 4  Results for different rating prediction algorithms with varying values of λ using the Amazon dataset.**

**Table 6  The rating scales for the different variants of the ratings.**

| Rating name | Amazon | Yelp |
|---|---|---|
| | **Rating scale** | |
| Explicit rating (overall rating) (*O*) | [1, 5] | [1, 5] |
| Implicit rating using sentiment words (*ImplicitRating_SW*) | [−9.612, 20.661] | [−11.171, 6.662] |
| Implicit rating using aspect (*ImplicitRating_Aspect*) | [−5.582, 15.618] | [−9.195, 7.257] |
| *O* + *ImplicitRating_SW* (*AVG_ImpSW*) | [−2.291, 10.179] | [−4.098, 6.129] |
| *O* + *ImplicitRating_Aspect* (*AVG_ImpAspect*) | [−4.306, 12.330] | [−5.085, 5.831] |

ways to produce such a transformation, such as the MinMaxScaler (*Bisong, 2019*), shown in Eq. (8)):

$$R = X + \left[ (Y - X) * (Z - \overline{X}) \right] / (\overline{Y} - \overline{X}) \tag{8}$$

where *R* isthe new rating, *X* is the minimum value of the implicit rating range, and *Y* is the maximum value of the implicit rating range. We set the new rating scale to be [1, 5].

$[\overline{X}, \overline{Y}]$ represents the explicit (standard) rating scale, as shown in Table 6. Finally, $Z$ is the rating that needs to be transformed from the original scale to the new scale.

### Parameter k

As mentioned earlier, $k$NN algorithms are among the algorithms used in this study (see Table 3). The parameter $k$ is required for the $k$NN algorithms, which is the number of neighbours used to calculate the predicted rating for a specific user. The value of this parameter should be carefully chosen to obtain effective results. We tested five values for $k$: 5, 10, 20, 30, 40, and 50, using explicit ratings to choose the optimal value for $k$. The Euclidean distance is used to measure the degree of similarity between users. Three metrics: MAE, MSE, and RMSE, are used to evaluate the rating prediction accuracy of each $k$NN algorithm with different $k$ values. Table 7 illustrates the results of these metrics for the Amazon dataset.

The values in bold represent the best results for each metric. As can be observed, three of the four algorithms with $k = 50$ produced the best results. Thus, the number of neighbours in the $k$NN rating prediction algorithms for the Amazon dataset was set to 50. Using the Yelp dataset, similar tests were done, and $k = 30$ gave the best results.

## RESULTS

This section presents the results for the conducted experiments involving the Amazon and Yelp datasets, with $\lambda = 20$ and $\lambda = 10$ respectively. The Amazon dataset contains 13,214 users, 17,022 movies, and 650,145 reviews, whereas the Yelp dataset has 11,831 users, 20,358 restaurants, and 162,723 reviews. The four proposed ratings have been rescaled to [1, 5], which is consistent with the explicit rating. For the $k$NN algorithms, $k = 50$ and $k = 30$ were used for the Amazon and Yelp datasets, respectively. In both datasets, the Euclidean distance was chosen to measure user similarity because it produced the best results after several experiments. For the $k$NN set of algorithms, we considered both the user-based and item-based CF algorithms. Each experiment employed a five-fold cross-validation procedure, dividing the dataset into 80% training and 20% testing for each fold.

Table 8 depicts the results of all experiments based on three predictive accuracy metrics: MAE, MSE, and RMSE. The results demonstrate that, for all 11 CF rating prediction algorithms, the proposed ratings outperform the baseline ratings (*i.e.*, the explicit ratings) in the three metrics. Notably, the implicit rating using sentiment words (ImpSW) performed better than all the other three proposed ratings. The results indicate that the ImpSW rating accurately captures the user's perception of the consumed item, which benefits the CF performance. Additionally, it demonstrates the efficiency of the method of sentiment word extraction and the appropriateness and accuracy of the scores assigned to each sentiment word.

The implicit rating using aspects (ImpAS) likewise yielded small error metric values and performed effectively in all rating prediction algorithms. Although the performance of ImpSW is better than ImpAS, the difference is marginal. One of the reasons why ImpSW surpassed ImpAS is because the used datasets are not specialized in aspect-based recommendations. On the other hand, each review includes a certain number of aspects

**Table 7  Performance metrics for the experiments using *k*NN algorithms for Amazon dataset.** The values in bold represent the best results for each metric.

| Algorithm | k | MAE | MSE | RMSE |
|---|---|---|---|---|
| *k*NNBasic | 5 | 0.8258 | 1.2637 | 1.1241 |
| | 10 | 0.8078 | 1.1800 | 1.0863 |
| | 20 | 0.8057 | 1.1543 | 1.0744 |
| | **30** | **0.8075** | **1.1522** | **1.0734** |
| | 40 | 0.8094 | 1.1531 | 1.0738 |
| | 50 | 0.8108 | 1.1546 | 1.0745 |
| *k*NNWithMeans | 5 | 0.7618 | 1.0828 | 1.0406 |
| | 10 | 0.7400 | 1.0203 | 1.0101 |
| | 20 | 0.7290 | 0.9921 | 0.9960 |
| | 30 | 0.7261 | 0.9858 | 0.9929 |
| | 40 | 0.7248 | 0.9836 | 0.9918 |
| | **50** | **0.7236** | **0.9815** | **0.9907** |
| *k*NNWithZScore | 5 | 0.7561 | 1.1048 | 1.0511 |
| | 10 | 0.7353 | 1.0363 | 1.0180 |
| | 20 | 0.7247 | 1.0058 | 1.0029 |
| | 30 | 0.7211 | 0.9983 | 0.9991 |
| | 40 | 0.7197 | 0.9950 | 0.9975 |
| | **50** | **0.7195** | **0.9951** | **0.9975** |
| *k*NNBaseline | 5 | 0.7605 | 1.0679 | 1.0334 |
| | 10 | 0.7397 | 1.0030 | 1.0015 |
| | 20 | 0.7321 | 0.9780 | 0.9889 |
| | 30 | 0.7300 | 0.9720 | 0.9859 |
| | 40 | 0.7295 | 0.9692 | 0.9845 |
| | **50** | **0.7293** | **0.9689** | **0.9843** |

that can correctly calculate the final score of the review. The co-clustering rating algorithm is the only algorithm for which the ImpAS ratings outperform the ImpSW ratings on all metrics. This implies that the ImpAS method performs better than the ImpSW method when using the user clustering technique. The findings of ImpAS also indicate that these ratings can successfully convey the user's perspective on items via the extracted aspects. In addition, the results of AVImpAS and AVImpSW demonstrate that the performance of the CF rating prediction algorithms using overall ratings is enhanced when combined with the implicit ratings.

## DISCUSSION

This study presented four methods for calculating ratings based on user reviews which are then evaluated in terms of their accuracy in different CF rating prediction algorithms. The first method sums all the sentiment scores/ratings for the sentiment terms mentioned in the review. The second method calculates implicit ratings based on a weighted aggregate of the user's opinions regarding different aspects of an item. Two other methods were provided for determining if the implicit rating alone is sufficient for supplying personalized information

**Table 8  Performance metrics for all the rating prediction algorithms using the proposed and explicit ratings.** The values in bold show the best results among the approaches for each dataset.

| Algorithm | Amazon dataset | | | | | Yelp dataset | | | | |
|---|---|---|---|---|---|---|---|---|---|---|
| | ImpSW | ImpAS | AVImpSW | AVImpAS | Explicit | ImpSW | ImpAS | AVImpSW | AVImpAS | Explicit |
| BaselineOnly | 0.0542 | 0.0769 | 0.1041 | 0.1491 | 0.7441 | 0.1091 | 0.1132 | 0.2481 | 0.2519 | 0.9919 |
| Co-clustering | 0.1673 | 0.1919 | 0.5769 | 0.5132 | 0.7251 | 0.1277 | 0.1216 | 0.2742 | 0.2736 | 1.0521 |
| kNNBaseline (Item-based) | 0.0542 | 0.0773 | 0.1033 | 0.1490 | 0.7152 | 0.1095 | 0.1137 | 0.2487 | 0.2526 | 0.9940 |
| kNNBaseline (User-based) | 0.0556 | 0.0798 | 0.1052 | 0.1516 | 0.7293 | 0.1096 | 0.1138 | 0.2488 | 0.2527 | 0.9947 |
| kNNBasic (Item-based) | 0.0565 | 0.0780 | 0.1107 | 0.1581 | 0.763 | 0.1117 | 0.1145 | 0.2613 | 0.2638 | 1.0536 |
| kNNBasic (User-based) | 0.0662 | 0.0924 | 0.1177 | 0.1677 | 0.8108 | 0.1120 | 0.1146 | 0.2615 | 0.2639 | 1.0541 |
| kNNWithMeans (Item-based) | 0.0581 | 0.0792 | 0.1045 | 0.1512 | 0.7172 | 0.1119 | 0.1147 | 0.2612 | 0.2637 | 1.0522 |
| kNNWithMeans (User-based) | 0.0558 | 0.0798 | 0.1072 | 0.1540 | 0.7236 | 0.1120 | 0.1148 | 0.2615 | 0.2640 | 1.0535 |
| kNNWithZScore (Item-based) | 0.0585 | 0.0791 | 0.1009 | 0.1501 | 0.7154 | 0.1120 | 0.1147 | 0.2612 | 0.2637 | 1.0524 |
| KNNWithZScore (User-based) | 0.0556 | 0.0778 | 0.1035 | 0.1503 | 0.7195 | 0.1119 | 0.1147 | 0.2613 | 0.2639 | 1.0530 |
| NMF | 0.0780 | 0.0818 | 0.1110 | 0.1551 | 0.7714 | 0.1149 | 0.1173 | 0.2634 | 0.2656 | 1.0550 |
| NormalPredictor | 0.1200 | 0.1391 | 0.1917 | 0.2688 | 1.1921 | 0.1747 | 0.1742 | 0.3673 | 0.3702 | 1.3286 |
| SlopeOne | 0.0724 | 0.0903 | 0.1153 | 0.1631 | 0.7363 | 0.1119 | 0.1148 | 0.2625 | 0.2652 | 1.0547 |
| SVD | 0.0772 | 0.0954 | 0.1177 | 0.1596 | 0.7311 | 0.1111 | 0.1150 | 0.2482 | 0.2521 | 0.9898 |
| SVDpp | 0.0584 | 0.0801 | 0.1054 | 0.1501 | 0.7149 | 0.1104 | 0.1147 | 0.2465 | 0.2507 | 0.9845 |
| Average across all algorithms | **0.0725** | 0.0933 | 0.1450 | 0.1861 | 0.7673 | **0.1167** | 0.1191 | 0.2650 | 0.2678 | 1.0509 |

MAE

| Algorithm | Amazon dataset | | | | | Yelp dataset | | | | |
|---|---|---|---|---|---|---|---|---|---|---|
| | ImpSW | ImpAS | AVImpSW | AVImpAS | Explicit | ImpSW | ImpAS | AVImpSW | AVImpAS | Explicit |
| BaselineOnly | 0.0075 | 0.0118 | 0.0190 | 0.0392 | 0.9464 | 0.0247 | 0.0248 | 0.0990 | 0.1010 | 1.4956 |
| Co-clustering | 0.0506 | 0.0575 | 0.3616 | 0.3118 | 0.9971 | 0.0377 | 0.0292 | 0.1203 | 0.1192 | 1.6264 |
| kNNBaseline (Item-based) | 0.0074 | 0.0120 | 0.0191 | 0.0396 | 0.9509 | 0.0249 | 0.0250 | 0.0995 | 0.1016 | 1.5040 |
| kNNBaseline (User-based) | 0.0077 | 0.0125 | 0.0197 | 0.0406 | 0.9689 | 0.0249 | 0.0250 | 0.0997 | 0.1017 | 1.5052 |
| kNNBasic (Item-based) | 0.0080 | 0.0124 | 0.0210 | 0.0448 | 1.0745 | 0.0260 | 0.0254 | 0.1071 | 0.1089 | 1.6287 |
| kNNBasic (User-based) | 0.0112 | 0.0168 | 0.0246 | 0.0497 | 1.1546 | 0.0260 | 0.0255 | 0.1073 | 0.1090 | 1.6293 |
| kNNWithMeans (Item-based) | 0.0079 | 0.0124 | 0.0194 | 0.0403 | 0.9630 | 0.0260 | 0.0255 | 0.1071 | 0.1089 | 1.6261 |
| kNNWithMeans (User-based) | 0.0075 | 0.0125 | 0.0203 | 0.0415 | 0.9815 | 0.0260 | 0.0255 | 0.1074 | 0.1092 | 1.6302 |
| kNNWithZScore (Item-based) | 0.0085 | 0.0125 | 0.0195 | 0.0403 | 0.9697 | 0.0260 | 0.0255 | 0.1072 | 0.1089 | 1.6265 |
| kNNWithZScore (User-based) | 0.0079 | 0.0122 | 0.0199 | 0.0410 | 0.9951 | 0.0260 | 0.0255 | 0.1073 | 0.1091 | 1.6291 |
| NMF | 0.0118 | 0.0132 | 0.0213 | 0.0415 | 1.0562 | 0.0276 | 0.0271 | 0.1091 | 0.1108 | 1.6327 |
| NormalPredictor | 0.0250 | 0.0330 | 0.0588 | 0.1155 | 2.3681 | 0.0517 | 0.0503 | 0.2136 | 0.2155 | 2.7985 |
| SlopeOne | 0.0126 | 0.0167 | 0.0236 | 0.0476 | 1.0084 | 0.0261 | 0.0257 | 0.1086 | 0.1107 | 1.6329 |
| SVD | 0.0118 | 0.0164 | 0.0235 | 0.0437 | 0.9533 | 0.0253 | 0.0254 | 0.0994 | 0.1014 | 1.4939 |
| SVDpp | 0.0078 | 0.0125 | 0.0194 | 0.0395 | 0.9363 | 0.0250 | 0.0252 | 0.0988 | 0.1009 | 1.4865 |
| Average across all algorithms | **0.0129** | 0.0176 | 0.0460 | 0.0651 | 1.0883 | 0.0283 | **0.0274** | 0.1128 | 0.1145 | 1.6630 |

MSE

| Algorithm | Amazon | | | | | Yelp | | | | |
|---|---|---|---|---|---|---|---|---|---|---|
| | ImpSW | ImpAS | AVImpSW | AVImpAS | Explicit | ImpSW | ImpAS | AVImpSW | AVImpAS | Explicit |
| BaselineOnly | 0.0863 | 0.1086 | 0.1379 | 0.1980 | 0.9728 | 0.1573 | 0.1574 | 0.3147 | 0.3179 | 1.2229 |
| Co-clustering | 0.2250 | 0.2398 | 0.6013 | 0.5584 | 0.9986 | 0.1937 | 0.1707 | 0.3467 | 0.3452 | 1.2753 |

**Table 8** (*continued*)

| Algorithm | Amazon | | | | | Yelp | | | | |
|---|---|---|---|---|---|---|---|---|---|---|
| | ImpSW | ImpAS | AVImpSW | AVImpAS | Explicit | ImpSW | ImpAS | AVImpSW | AVImpAS | Explicit |
| *k*NNBaseline (Item-based) | 0.0861 | 0.1094 | 0.1382 | 0.1989 | 0.9752 | 0.1578 | 0.1580 | 0.3155 | 0.3188 | 1.2264 |
| *k*NNBaseline (User-based) | 0.0880 | 0.1120 | 0.1402 | 0.2014 | 0.9843 | 0.1579 | 0.1581 | 0.3157 | 0.3189 | 1.2268 |
| *k*NNBasic (Item-based) | 0.0896 | 0.1113 | 0.1449 | 0.2116 | 1.0366 | 0.1611 | 0.1594 | 0.3273 | 0.3300 | 1.2762 |
| *k*NNBasic (User-based) | 0.1056 | 0.1294 | 0.1570 | 0.2229 | 1.0745 | 0.1613 | 0.1595 | 0.3275 | 0.3302 | 1.2764 |
| *k*NNWithMeans (Item-based) | 0.0892 | 0.1111 | 0.1392 | 0.2008 | 0.9813 | 0.1612 | 0.1596 | 0.3273 | 0.3300 | 1.2752 |
| *k*NNWithMeans (User-based) | 0.0868 | 0.1119 | 0.1423 | 0.2038 | 0.9907 | 0.1613 | 0.1597 | 0.3277 | 0.3305 | 1.2768 |
| *k*NNWithZScore (Item-based) | 0.0921 | 0.1120 | 0.1396 | 0.2007 | 0.9847 | 0.1612 | 0.1596 | 0.3274 | 0.3301 | 1.2753 |
| *k*NNWithZScore (User-based) | 0.0888 | 0.1103 | 0.1410 | 0.2025 | 0.9975 | 0.1612 | 0.1597 | 0.3276 | 0.3304 | 1.2771 |
| NMF | 0.1085 | 0.1150 | 0.1458 | 0.2038 | 1.0277 | 0.1662 | 0.1645 | 0.3303 | 0.3328 | 1.2777 |
| NormalPredictor | 0.1581 | 0.1816 | 0.2425 | 0.3399 | 1.5389 | 0.2274 | 0.2243 | 0.4622 | 0.4643 | 1.6729 |
| SlopeOne | 0.1123 | 0.1292 | 0.1537 | 0.2183 | 1.0042 | 0.1614 | 0.1604 | 0.3296 | 0.3327 | 1.2782 |
| SVD | 0.1086 | 0.1281 | 0.1532 | 0.2091 | 0.9764 | 0.1591 | 0.1593 | 0.3152 | 0.3185 | 1.2222 |
| SVDpp | 0.0884 | 0.1117 | 0.1393 | 0.1986 | 0.9676 | 0.1581 | 0.1587 | 0.3142 | 0.3176 | 1.2192 |
| **Average across all algorithms** | **0.1076** | 0.1281 | 0.1811 | 0.2379 | 1.0341 | 0.1671 | **0.1646** | 0.3339 | 0.3365 | 1.2852 |
| | | | | | RMSE | | | | | |

that may be utilized effectively to improve CF rating prediction algorithms or whether the explicit rating should be combined with the implicit ratings. These ratings were integrated and experimented into various CF rating prediction algorithms, and the results are shown in Table 8.

The results show that the proposed methods obtained good rating prediction accuracy compared to the baseline rating, with the ImpSW approach giving the best average results across all algorithms for the Amazon dataset and the ImpSW giving the best results for the Yelp dataset (except for the MAE metric). This might be due to the nature of the used datasets. The Yelp dataset may be more suited for aspect-based CF because each review contains a variety of aspects that may assist the ImpAS rating to be more accurate than the ImpSW rating.

The results also show that the combinations of explicit and implicit ratings enhance the accuracy of all the tested CF prediction algorithms compared to using the explicit ratings alone. Our interpretation of this increment is that an explicit rating is a numerical evaluation of an item using a particular scale that expresses the user's general opinion. It cannot convey a fine-grained understanding of the underlying assumptions driving user ratings. It merely expresses the coarse-grained rating and cannot capture the specific user preferences or interests in each part of the item to comprehend user opinions and analyze user behaviour. For instance, just because a user gives an item a high rating does not necessarily imply that he likes the item as its whole. He might still dislike some particular features of that item. Also, a negative rating does not necessarily mean the user dislikes everything about the item. Contrarily, implicit ratings are derived from user reviews. The user will only express an opinion on an item's features that excite his attention, making this rating more accurate because it comes from the user's specific information. This, in turn,

aids in understanding fine-grained opinion mining versus coarse-grained, particularly when using the aspects.

To summarize, user reviews provide more detailed and qualitative information compared to numerical ratings. They allow users to express their opinions, experiences, and insights about a particular item or service. This rich textual content offers a deeper understanding of user preferences, factors influencing their choices, and the specific aspects they liked or disliked. Such detailed information can be valuable for generating personalized recommendations that align with user preferences. Adding the implicit rating (specific opinion) to the explicit rating (general opinion) will improve the understanding of the user preferences, yielding to increase in the performance of the CF rating prediction algorithms.

From a different perspective, the results show that ratings solely derived from user reviews are sufficient for reliable prediction accuracy. For instance, the average MAEs for all algorithms on the Amazon dataset are 0.0725 and 0.0933 for the ImpSW and ImpAS, respectively, compared to 0.1451 and 0.1861 for the AVImpSW and AVImpAS, and the same is true for the Yelp dataset. Since the average MAEs using implicit ratings exceeded the average of MAEs combining implicit and explicit ratings, adding general information to specific information cannot always boost the CF performance. In contrast, the opposite is true, as described before.

It is interesting to see the four different types of $k$NN algorithms used during the experiments, with user-based and item-based methods implemented for all algorithms. The results show that the item-based method outperformed the user-based method for all four types. This is owing to the characteristics of the dataset, which includes a larger number of items than users in both datasets, thus making the item-based method more suitable and providing superior performance.

Overall, the four proposed ratings—ImpAS, ImpSW, AVImpAS, and AVImpSW—affected the accuracy of CF rating prediction algorithms, which is evidenced by the three error metrics. The results demonstrate that the ratings derived from user reviews using either sentiment words or aspect—sentiment word pairs are indeed better options to reflect the user's impression of the consumed item. The results further suggest the potential of using implicit ratings instead of explicit ratings in CF recommendation systems.

## CONCLUSION AND FUTURE WORKS

This study explored review-based CF approaches that use different ratings extracted from user reviews. It centred on calculating the implicit rating to take advantage of the wealth of information in user reviews to improve the CF performance. These ratings were being experimented in various CF rating prediction algorithms. To our knowledge, it is the first study to demonstrate different methods for generating implicit ratings on large-scale datasets.

Specifically, this study comprised three main tasks: the first was to calculate two variants of the implicit rating derived from user reviews. The first implicit rating was calculated using sentiment words by aggregating the scores of all extracted sentiment words mentioned in a particular review. While the second implicit rating was based on aspect—sentiment word

pairs and calculated using the weighted sum of the user's opinions about multiple aspects. The second task produced two additional ratings by independently combining the explicit and implicit ratings generated in the previous task. In the last task, the generated ratings are independently incorporated into various CF rating prediction algorithms to evaluate their accuracy compared to the explicit ratings.

Experiments were conducted on two real large-scale datasets: Amazon and Yelp, to determine the impact of the generated ratings on several CF rating prediction algorithms. Results show that the proposed ratings enhanced the accuracy of CF rating prediction algorithms and outperformed the explicit ratings in terms of three predictive accuracy metrics. Moreover, the findings demonstrate that implicit rating accurately captures the user's opinion of the consumed item.

The near future works include extracting more information from the user-generated reviews, such as contextual information and reviewers' emotional elements, to better understand user preferences and improve the CF performance. Additional possible future studies may involve expanding the presented methodologies to cross-domain and group recommender systems and investigating how user reviews may be utilized to support explain ability in recommender systems.

### Funding
This work was under the funding of the Universiti Kebangsaan Malaysia with the grant number: DIP-2020-017. The funders had no role in study design, data collection and analysis, decision to publish, or preparation of the manuscript.

### Grant Disclosures
The following grant information was disclosed by the authors:
The Universiti Kebangsaan Malaysia: DIP-2020-017.

### Competing Interests
The authors declare that there are no competing interests.

### Author Contributions
- Sumaia AL-Ghuribi conceived and designed the experiments, performed the experiments, analyzed the data, performed the computation work, prepared figures and/or tables, authored or reviewed drafts of the article, and approved the final draft.
- Shahrul Azman Mohd Noah conceived and designed the experiments, analyzed the data, prepared figures and/or tables, authored or reviewed drafts of the article, and approved the final draft.
- Mawal Mohammed analyzed the data, authored or reviewed drafts of the article, and approved the final draft.

### Data Deposition
Code is available Supplemental Files.

The Amazon Dataset collected by Julian McAuley and is available at UC San Diego: https://cseweb.ucsd.edu/~jmcauley/datasets.html#amazon_reviews

The Yelp dataset is available at Yelp: https://www.yelp.com/dataset

## Supplemental Information

Supplemental information for this article can be found online at http://dx.doi.org/10.7717/peerj-cs.1525#supplemental-information.

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
