# Peer review of "An experimental study on the performance of collaborative filtering based on user reviews for large-scale datasets"

_PeerJ Computer Science, doi:10.7717/peerj-cs.1525_

## Round 0.1 · original submission · Major Revisions

Authors should carefully revise the manuscript based on the reviewers' comments. Especially it should be compared with recent methods, as mentioned by reviewer 1. Reviewer 2 also thinks that some of the descriptions in the paper are confusing, and the author should respond to his comments in great detail and make careful revisions. Although both reviewers suggested major revisions to the paper, their comments pointed out obvious flaws. It is hoped that the authors will carefully consider their comments and further improve the manuscript.

Reviewer 1 ·

Basic reporting

This article details the important role of collaborative filtering algorithms in certain contexts and points out that implicit evaluation should also be carefully considered. And most existing algorithms ignore this one. This work proposes four methods to compute implicit ratings in user reviews. The ratings extracted from user comments are based on degree and aspect sentiment word pairs, and experiments are done on two large-scale datasets to demonstrate their superiority.

Experimental design

The corresponding experiments were designed on two publicly available datasets in this work, and the experimental results were superior. However, the selected baseline is less, especially the relevant research methods in the last two years. Besides, this paper also lacks the corresponding ablation experiments to investigate the importance and practical contribution of different components in this work.
Therefore, the authors need to add relevant baselines to make the experimental results more convincing, and also need to add relevant ablation experiments to facilitate the reader's understanding of this work.

Validity of the findings

This work illustrates the importance of implicit scores by experiments and gives the corresponding algorithm, but it lacks the expression of the main contribution to the work. For example, there is only one sentence in the Abstract expressing the superiority of this work.
To address the above issues, there is a need to strengthen the presentation of the significance of this work and supplement the conclusion and abstract.

Additional comments

The spacing of the bars in Figure 3 is inconsistent and probably needs to be improved to be more aesthetically pleasing.

Reviewer 2 ·

Basic reporting

The sections before "Experimental Results and Analysis" is well-written, and the others should be improved. For example, the overall tense in “The Dataset” section is confusing.

Experimental design

The implicit rating using sentiment words in Task 1 (Line 261) is based on the work [45]. But the authors regard it as their own method. Could the author briefly explain how it differs from [45] in the part of Task 1?

Validity of the findings

1. In Line 486, the authors claim that “the need to combine the implicit ratings with the explicit ratings is not necessary”. This is indeed demonstrated in the experimental results. However, this is a counter-intuitive conclusion, because combining information from different sources at the same time usually gets a better result. I think the authors need to make some theoretical arguments for the results.
2. In Line 387, the authors show the results for different lambda values, and claim that “the prediction accuracies for varying lambda values are not significantly different”. However, as seen in Figure 4(a)(c)(e)(f), there is still a significant variation over the MSE metrics. Furthermore, according to the actual situation, the larger the lambda value, the more valid the rating of the user/item is. If the authors believe that lambda has little impact, I suggest choosing the largest lambda value on both datasets.

Additional comments

No comment.

---

## Round 0.2 · accepted · Accept

All reviewers think that the author's response solved their concerns, and the author further improved the quality of the paper based on the reviewer's comments. Now I recommend accepting this paper for publication in PeerJ Computer Science.

Reviewer 1 ·

Basic reporting

This paper details the important role of collaborative filtering algorithms in certain situations and points out that implicit evaluation should also be carefully considered. And most of the existing algorithms ignore this one. This work proposes four methods to compute implicit evaluations in user reviews. The evaluations extracted from user reviews are based on degree and aspect-based sentiment word pairs, and experiments are done on two large-scale datasets to demonstrate their superiority.

Experimental design

In this work, corresponding experiments were designed on two publicly available datasets with superior results. The updates are detailed to show that this algorithm has performance under comparative experiments. The usefulness of the method is highlighted

Validity of the findings

This work illustrates the importance of implicit scores through experiments and gives the corresponding algorithm that complements the conclusions and abstract well.

Reviewer 2 ·

Basic reporting

The authors have solved all my problems in the last revision phase.

Experimental design

The authors have solved all my problems in the last revision phase.

Validity of the findings

The authors have solved all my problems in the last revision phase.

Additional comments

The authors have solved all my problems in the last revision phase.